# DNA-Assisted Assembly of Gold Nanostructures and Their Induced Optical Properties

**DOI:** 10.3390/nano8120994

**Published:** 2018-12-01

**Authors:** Jiemei Ou, Huijun Tan, Xudong Chen, Zhong Chen

**Affiliations:** 1Key Laboratory for Polymeric Composite and Functional Materials of Ministry of Education, Guangdong Engineering Technology Research Center for High-Performance Organic and Polymer Photoelectric Functional Films, School of Chemistry, Sun Yat-sen University, Guangzhou 510275, China; oujiemei12@163.com (J.O.); tanhj5@mail2.sysu.edu.cn (H.T.); cescxd@mail.sysu.edu.cn (X.C.); 2National Engineering Research Center for Healthcare Devices, Guangdong Key Lab of Medical Electronic Instruments and Polymer Material Products, Guangdong Institute of Medical Instruments, Guangzhou 510500, China

**Keywords:** gold nanocrystals, programmability, plasmonic chirality, SEF, SERS

## Abstract

Gold nanocrystals have attracted considerable attention due to their excellent physical and chemical properties and their extensive applications in plasmonics, spectroscopy, biological detection, and nanoelectronics. Gold nanoparticles are able to be readily modified and arranged with DNA materials and protein molecules, as well as viruses. Particularly DNA materials with the advantages endowed by programmability, stability, specificity, and the capability to adapt to functionalization, have become the most promising candidates that are widely utilized for building plenty of discrete gold nanoarchitectures. This review highlights recent advances on the DNA-based assembly of gold nanostructures and especially emphasizes their resulted superior optical properties and principles, including plasmonic extinction, plasmonic chirality, surface enhanced fluorescence (SEF), and surface-enhanced Raman scattering (SERS).

## 1. Introduction

Nanoparticles (NPs) as a kind of “artificial atoms” have become a new part of chemistry since the 1990s. This kind of nanomaterials shows remarkable physical and chemical properties which could provide great technological breakthroughs in electronic, catalytic, and sensing applications [1] as well as in spectroscopy if the geometric morphology and arrangement of the particles can be finely tailored. Gold nanoparticles (AuNPs) especially have drawn much attention during the past few decades due to its merits of simple operation, high stability, and ease of synthesis. Nowadays, varied morphological AuNPs such as nanospheres [2,3,4,5,6,7], nanorods [8,9,10,11,12], nanocubes [13], dendrimers [14], nanodisks [15], hexagons [16], nanostars [17], nanopyramids [14], and nanoflowers [18] have been successfully synthesized and exhibit typical optical properties. However, it remains a great challenge to spatially organize these AuNPs with a precisely controllable distance and orientation for investigating the electronic and optical coupling between the NPs or for their application in plasmonic devices. Many existing industrial techniques to arrange AuNPs into well-aligned nanostructures have been developed, including electron beam lithography [19,20], Langmuir-Blodgett techniques [21], and electrochemical deposition [22], all of which are laborious and costly. Other chemical methods like covalent linkage and electrostatic coupling [23] suffer from a lack of repeatability and binding specificity. So, it is of significance to exploit new techniques to efficiently and promptly assemble NPs. DNA double helix structures with specific base-pairing interactions existing between the A-T and G-C base pairs were firstly revealed by Watson and Crick as early as 1953 [24]. Till the 1980s, Seeman initially took advantage of the high specificity of DNA materials as a chemical material in the construction of nanoscales [25]. From then on, the emerging structural DNA nanotechnology as a powerful route exhibits a tremendous superiority used for self-assemble biocompatible materials in a fully programmable, sequence-encoded fashion.

AuNPs are easy to modify with oligonucleotides via the stable thiol-gold chemical bond and the resulting DNA-AuNPs can be rationally assembled by DNA hybridization. Once AuNPs are arranged into well-defined configurations, they will exhibit distinctive and substantial optical properties different from the bulk and single particles. In 1996, brand-new DNA-AuNPs composites were simultaneously exploited by Mirkin [26] and Alivisatos [27], which opened a wide avenue for the massive self-assembly of AuNPs complexes. Most importantly, Rothemund [28] and Douglas [29] successively designed 2D and 3D DNA origami tiles from 2006 to 2009, and the pioneering works in the field of structural DNA nanotechnology have been used as a versatile and robust platform for precisely constructing plenty of sophisticated AuNPs patterns. So far, many preceding reviews mainly focused on the development of DNA nanotechnology and their applications in nanoparticle self-assembly [30,31,32,33,34,35]. However, a systematical and comprehensive investigation of how the varieties of gold nanostructures matter with their special optical properties is yet to be summarized in detail. Therefore, in this review, we strategically concluded the previously mentioned excellent work on DNA addressable nanostructures, ranging from one- three-dimensional patterns, from isotropic to anisotropic, as well as the shape of the AuNPs range from nanospheres to nanorods, nanotriangles, and nanostars, concurrently focusing on the gold nanostructure-related local-electromagnetic-field-sensitive optical activities, which are sensitive to the interparticle gap, spatial arrangement, and orientation of NPs, including plasmonic extinction, plasmonic chirality, surface enhanced fluorescence, and surface-enhanced Raman scattering (Scheme 1).

## 2. Structural DNA Self-Assembly Nanotechnology

Bottom-up structural DNA self-assembly nanotechnology makes use of DNA strands to build highly programmable supramolecular structures. Such a research field has been experiencing a fruitful development over the past three decades. In general, structural DNA nanotechnology can be traced back to the innovation by Nadrian Seeman in the 1980s, who began to use DNA as a chemical material for the construction of relatively flexible branched junction structures [36] and topological structures [37]. The emerging DNA nanotechnologies as promising artificial molecules have grown rapidly and provided a useful technological platform suitable to exhibit the special role of DNA in the nanomaterial sciences. The fabrication of crossover DNA tiles [38,39,40] is an important but transitional period in the area of DNA self-assembly nanotechnology. These tiles with greater rigidity could be used to assemble higher-order periodic and aperiodic lattices [41,42,43,44]. Soon afterward, the most significant breakthrough in the field of DNA self-assembly was the invention of the DNA origami technique designed by Rothemund in 2006 [28], which was a milestone advancement in constructing planar, arbitrarily-shaped, two-dimensional (2D) object DNA nanostructures. To be specific, a long scaffold strand (single-stranded DNA from the M13 phage genome, ~7429 nucleotides long) was folded into prescribed shapes by a pool of ~200 short staple strands through sequence-specific hybridization and the formation of multiple crossovers. Later, Shih et al. extended this strategy from 2D planar origami structures to arbitrary 3D nanoarchitectures in 2009 [29]. In contrast to early DNA tiles assisted by an exact stoichiometry and highly purified DNA sequences, DNA origami can produce well-formed 2D and 3D structures in near-quantitative yields with unpurified oligonucleotides, which largely simplifies the experimental design and extends their further applications. The sequence specificity and the resulting spatial addressability of DNA nanostructures are especially well suited for the arrangement of nanoparticles. Based on this advantage, the following sections will introduce in detail the engineerable nanoarchitectures and their induced optical properties through DNA-mediated self-assembly.

## 3. The DNA-Assisted Self-Assembly of AuNP Nanoarchitectures

Oligonucleotides are typical of short single-strand DNA molecules, oligomers, using solid-phase chemical synthesis with a user-specified sequence. Since Alivisatos [27] and Mirkin [26] pioneered the oligonucleotides-templated AuNP nanostructures via complementary base-pair interaction after the surface modification of gold nanoparticles, it has rapidly attracted much attention in assembling multicomponent frameworks and largely extended their applications in sensing or optoelectronics. In the primary stage, linear AuNPs dimeric or trimeric conjugates were attained through directly modifying the AuNPs with thiol-terminated oligonucleotides, which soon became the most general strategy, widely used in research on the DNA-based self-assembly of AuNPs. This technique can precisely control the interparticle distance scaled down to several nanometers for studying the strong plasmon coupling between closely spaced AuNPs. For examples, Bidault et al. [45] demonstrated the synthesis of well-defined AuNPs dimers and trimers with nanometer spacing obtained by hybridizing mono-conjugated DNA-particle building blocks. To bring the particle spacing down to the nanometer range, the authors functionalized 8 nm AuNP with a single 100-base single-stranded 5′ thiolated DNA molecule (building blocks **1**) and sequentially functionalized it with a 5′ thiolated 50 base single-stranded DNA molecule to yield building block **2**. The hybridization of building block **1** or building block **2** with 5-nm and 18-nm diameter AuNPs monofunctionalized with a 3′ thiolated complementary strand yielded structures **3**, **4**, and **5** (Figure 1A). Further, Busson et al. [46] used larger AuNPs, 36 nm in diameter, to assemble symmetric or asymmetric high purity AuNPs dimers with substantial scattering cross sections and plasmon coupling using either a 50-bp (base pair) or 30-bp DNA linker perpendicular to (or parallel to) the dimer axis and the distance between the dimer can be tuned to be as short as 7 nm (Figure 1B). Liu’s group [47] extended the self-assembly behavior to successfully prepare DNA bimodified GNPs (gold nanoparticles) at different ends. It is well-known that the duplex DNA sequence is hydrogen-bond-dependent and sensitive to its surroundings, even folding itself to form a loop structure via the base-pair complementarity. Accordingly, Lermusiaux et al. [48] introduced simple dynamic DNA-AuNP dimers linked by a single stem-loop containing DNA sequence varieties that are reversibly through the hybridizing or removal of a single target DNA strand. Specifically, sequence **S** is designed to contain a 10 bp stem-loop which opens when hybridizing a **T** single-strand, and the 100 bases-long **S** sequence can be hybridized to a complementary **C** DNA molecule that is over 50 bp. Structures **1**, **2**, and **3** in Figure 1C are obtained via the electrophoresis separation of 8-nm AuNPs linked to a single thiolated **S**, **C** or **S** + **T** DNA molecule. The closed **4** and open **5** dimers are obtained by hybridizing structures **1** + **2** and **2** + **3**, respectively (Figure 1C). Besides those simpler dimeric or trimeric nanoassemblies, complex nanostructures are also realized with an ingenious design. For instance, Sleiman’s group [49] developed a straightforward strategy to stably anchor one or more AuNPs to either the interior or terminal position in the DNA strand, forming well-defined AuNP squares and rectangles using cyclic disulfide: (i) AuNPs were modified internally with cyclic disulfide **D**, (ii) the addition of an extension strand, (iii) electrophoresis purification of the DNA-AuNP conjugate by a displacement strand, (iv) the tetramer structure **I** attained by annealing four such monoconjugates **Au-1_a-d_**. Similarly, the linear tetrameric structure **II** is also prepared by the selective functionalization of the **D**-modified strands **2** (step i) and **3** (step ii) with two different sizes of nanoparticles. This strategy has the potential to enable the construction of any number of discrete DNA-AuNP conjugates (Figure 1D).

However, the linker DNA molecule and duplex DNA, which served as the bridge, are not sufficiently rigid such that the arrangement of AuNPs constantly suffers from deformation. To tackle this problem, DNA origami with a structural rigidity has been proven to be a robust scaffold that is broadly applied when building well-defined nanostructures. Yan Hao’s group has aimed to study DNA origami-based self-assembly and they have constructed plenty of different types of particle patterns since 2008. Sharma et al. [50] employed an activated lipoic ester that reacted with amine-modified-DNA to generate dithiol-modified DNA mixed with 10-nm AuNPs. The resulting AuNP-DNA conjugates hybridized with a single-stranded viral M13 DNA showed that the yield of the origami tile-templated AuNP structures was 45% (the monothiol approach) and 91% (the dithiol approach) (Figure 2A). Ding et al. [51] demonstrated a different efficient approach for the bowtie-like self-similar chain alignment, in which the 15-, 10-, and 5-nm diameter AuNPs individually modified with multiple short DNA strands were prepared for mixture with a triangular DNA origami structure containing three captured strands at each immobilized site for the AuNPs selective attachment (Figure 2B). Aside from the isotropic spherical AuNPs, Yan’s group [52] also developed anisotropic gold nanorods (AuNRs) attached to the arms of triangular DNA origami (Figure 2C). CTAB-capped AuNRs were coated with a thin gold layer via overgrowth for binding to thiol-DNA and the resultant DNA-AuNR conjugates were immobilized onto two arms of the DNA tiles to form an adjustable angle, ranging from 0° (side by side arrangement) to 60°, 90°, and 180° (end-to-end arrangement). Moreover, Wang’s group [53,54] recently combine AuNPs and AuNRs to construct a series of well-defined heterostructures by using DNA origami clamps (Figure 2D), which greatly expanded the utility of DNA origami as a tool for building more complex nanoarchitectures. Inspired from the preceding studies in the literature, the strongly plasmonic couples always happens in the vicinity of the sharp tips [55], and many researchers have been dedicated to rationally design and fabricate sharp tips containing metallic nanostructures. Sen’s group [56] and Ding’s group [57], respectively, organized representative Au nanostars and Au nanoprisms into sharp tips containing gold nanoantennas with an amplified local electromagnetic field in a single-molecule SERS study. These two works will be discussed in detail in the following SERS sections.

## 4. The Plasmonic Extinction of DNA-Assisted Self-Assembly of AuNP Nanoarchitectures

Plasmons are a kind of collective oscillation in the electron density of metals or semiconductors induced from light. A restoring force arises from the Coulomb interaction between electrons and nuclei that leads to the oscillation of the electron cloud relative to the nuclear framework when the electron cloud is displaced relative to the nuclei (Figure 3A). The localized surface plasmon resonance near-field coupling between surface plasmons of neighboring particles leads to energy shifts and energy confinements between particles [58,59,60]. It is well-known theoretically [61] and observed experimentally [62] that once two nanoparticles are brought into proximity, their plasmons couple shifts the resonance wavelength depending on the particle separation: the localized surface plasmon resonance peak becomes significantly red-shifted due to the short particle spacing because of the electromagnetic retardation and the strong near-field plasmon coupling [63]. Otherwise, the plasmon resonance distributions blue-shift with an increasing interparticle distance due to reducing the near-field coupling (Figure 3B). When the nanostructures are much smaller than the wavelength, the plasmonic effects can be used to manipulate light through the nanometer-sized metal nanoparticles. Hence, plasmonic extinction spectra characterization has been commonly studied around all of the nanoassemblies. Structural DNA nanotechnology provides a practical route to self-assemble and precisely control the placement of metal nanoparticles for plasmonic research. Dimer nanostructures (as the simplest nanosystem) are easily achieved by this technique. In 2013, Wang’s group [64] reported the self-assembly of symmetric and asymmetric AuNP dimer structures with tunable sizes and interparticle distances via the DNA self-assembly strategy (Figure 4A). They systemically investigated the size- (13, 20, and 40 nm) and distance-dependent (5, 10, and 15 nm) plasmon coupling of ensembles of single and isolated dimers in solution, showing an expected red-shift with the increasing particle sizes or reducing interparticle distances, in agreement with the theoretical calculations. The aforementioned DNA origami scaffolds were also utilized to control the position and relative angle of the two AuNRs created by Yan’s group [52]. The AuNRs-involved four different relative angles that were designed from 0° to 60°, 90°, and 180°. The researchers observed a red-shift of the LSPR peak for constructs 180° and 60° by ~9 and ~6 nm, respectively, and a ~5.5 nm blue shift for construct 0° and almost no shift for construct 90° (Figure 4B). All of those experimental results are also coincide with the theoretical simulations of the optical spectra corresponding to each of the AuNR dimer constructs. On account of the achievements above, Wang’s group [54] extended the homogeneous assembly of gold nanoparticles into heterogeneous plasmonic nanostructures by using the DNA origami clamps and studied the plasma coupling among the assembled plasmonic nanostructures (Figure 4C). Interestingly, they discovered that only when two or more AuNRs were closely arranged, can the longitudinal resonance peak can show red-shifts to some extent. Otherwise, the longitudinal resonance peak of the AuNRs does not red-shift even when they were separated by small spherical AuNPs, which further elucidates that plasmon coupling is appreciably sensitive to the adjacent nanoparticles.

## 5. The Plasmonic Chirality of DNA-Directed AuNP Nanoarchitectures

Although plasmonic extinction can show the relationship between nanostructures and plasmon coupling to some extent, it still remains elusive in validating the more specific architectural features, especially whether plasmonic extinction has any effect on the enantiomeric structures. The plasmonic chiroptical effect exhibits a characteristic bisignated CD (circular dichroism) spectrum that provides more detailed information for specifically studying the mechanism of plasmon coupling. Artificially engineered DNA nanotechnology shows a great potency in the fabrication of plasmonic chiral nanostructures that generally display exceptionally strong optical activity and circular dichroism, which have also been generalizable to their widespread potential applications in negative refractive index media [65], broad-band circular polarizers [66], and ultrasensitive sensing devices [67,68]. Plasmon–plasmon couplings and exciton-plasmon interaction are the different physical mechanisms for generating a chiral response from DNA-AuNP structures. Two representative physical principles were previously demonstrated by Wang’s group, shown in Figure 5. The first one was termed plasmon hybridization and depended on the arrangement of AuNPs analogous to molecular orbital hybridization, in which the DNA molecule was only treated as the template for guiding the structural assembly but was not thought of as a passive structural element. The other one renders the transfer of the molecular chirality to the plasmonic nanostructures due to the enhanced electromagnetic field, caused by the chiral molecule-involved dipole–plasmon Coulomb interactions. In this review, we purposely focus on the nanoassemblies guided by DNA but not the DNA molecule itself, so we only pay attention to the first plasmon–plasmon coupling nanostructures in the following section.

As early as 2009, Paul Alivisatos [70] elaborately designed and created discrete chiral pyramids by immobilizing four different sizes of gold nanocrystals at the tips and building a structure with inverted symmetry by switching the placement of any two nanocrystals. This is the first time to obtain artificial plasmonic enantiomeric assemblies through the greater physical arrangement of metallic nanostructures. Later, aided by the emerging DNA origami technique, the exquisite design and exact stoichiometric ratio of DNA sequences were no longer necessary in the assembling of plasmonic systems, which enabled researchers to greatly simplify the plasmonic chiral arrangements and regulate the number of particles. In 2012, Ding’s group [71] demonstrated the construction of 3D plasmonic chiral nanostructures by rationally rolling and stapling AuNP-dressed rectangular origami sheets (Figure 6A). They found that the coupling strength between the AuNPs could be increased through the enlargement of AuNPs and they attained fully engineerable plasmonic chiral nanomaterials. The authors subsequently further shortened the multiple annealing process by directly organizing four nominally identical DNA modified AuNPs onto a rigid addressable rectangle origami template, forming a three-dimensional asymmetric tetramer [72]. The obtained left- and right-handed assemblies present a characteristic bisignate peak-dip CD shape and a bisignate dip-peak CD profile, respectively, agreeing well with the theoretical predictions (Figure 6B). With this strategy, Dai et al. [73] successfully arranged a series of three-dimensional AuNPs tetrahedron nanoarchitectures by tuning different particle sizes and interparticle distances, which indicated that the plasmonic resonance coupling of the AuNPs tetramer is proportional to the particle size and inversely proportional to the interparticle distance. Another systematic investigation of the chiral plasmonic nanostructures with spherical AuNPs was explored by Tim Liedl’s group [74], who created perfect assemblies of right- and left-handed helices with nine nanoparticles using DNA origami 24-helix bundles (Figure 6C). The researchers tailored the geometrical and material parameters using simple adjustments of the design and fabrication protocols and the experimentally observed CD spectra agreed well with theoretical calculations based on classical electrodynamics. 

Benefited from DNA origami technology, the orientation, and arrangement of anisotropic AuNRs can be precisely controlled into well-defined spatial configurations and geometries [75,76,77,78,79,80,81,82,83] which are difficult to obtain by simple DNA linkers or other biomaterial molecules. In 2013, Wang’s group [75] first reported the examples of discrete 3D anisotropic building block nanoassemblies by precisely assembling the AuNRs on opposite sides of bifacial DNA origami and experimentally observing distinct plasmonic chiral responses by rationally manipulating the location of the AuNRs on the origami template (Figure 6D). Then, more complicated AuNR helical superstructures were also constructed by Lan and co-workers [76]. The two helices contained 9 AuNRs helical superstructures with right and left-handedness, respectively (Figure 6E). The resulting plasmonic chiral responses were delicately tuned by the number of AuNRs contained in the helices. Depending on this work, Lan et al. [77] continued presenting a versatile DNA origami adapter that could programmatically self-assemble into novel stair-like and coil-like AuNR chiral supramolecular architectures, exhibiting distinctive chiral optical responses (Figure 6F). Recently, the authors went further by dynamically controlling an AuNR 3D chiral plasmonic helix with a fully switchable chirality by using DNA-toehold-mediated conformational changes in the DNA template, which enhanced the understanding of the physical phenomena of magnetochirality and chiral quantum optics [81]. Similar to the heterogeneous AuNPs tetrahedron with different sizes and interparticle distances mentioned above, AuNRs assemblies can also be readily tuned by size, number, and spatial configuration. Shen et al. [80] successfully fabricated a series of chiral heterogeneous AuNR dimers and trimers of different sizes and numbers and examined their chiroptical responses. Additionally, Chen et al. [83] intelligently produced a series of heterogeneous AuNR trimers and their daughter AuNR dimers to investigate the origin of the plasmonic chirality of AuNR trimers. The authors experimentally corroborated using CD spectroscopy that the plasmonic chirality of the three T-shaped plasmonic AuNR trimers is a nearly perfect summation of the chiroptical response of all their possible dimeric components, independent of the handedness and the amplitude of the CD signals, which provided fundamental insights into the nature of the optical chirality at the nanoscale and established generalizable design rules for the next-generation DNA-origami-templated nanodevices with predictable and tailorable chiral responses. To raise the horizon of the bottom-up construction of other nanostructures with growing complexity, great progress in AuNR and AuNP plasmonic composites has also been achieved by DNA origami, which is difficult to complete with conventional methods. Shen et al. [53] successfully developed an effective strategy to assemble a novel set of heterogeneous AuNR@AuNP plasmonic helices by rationally designing the chiral patterning of eight DNA recognition sites in a helical arrangement around an AuNR and attaching AuNPs to the prearranged sites on the AuNR surface. The resulted complex plasmon-plasmon coupling between the AuNR and AuNPs helix could generate their unique chiroptical activities.

## 6. The Surface-Enhanced Fluorescence of DNA-Directed AuNP Nanoarchitectures

Fluorescence originates from the energy transfer between the excited state and ground state, leading to the emission of light by a substance that has absorbed light or other electromagnetic radiation. Based on plasmon-exciton coupling, fluorescence is abundantly tailored in quenching [84,85,86], enhancement [87,88,89], emission directionality [90,91], photobleaching reduction [92,93], or in the modulation of the emission spectral profile [52,94,95]. Among these, surface-enhanced fluorescence (SEF) has attracted appreciable interest because it can facilitate the scrutiny of single-molecule fluorescence. In 2006, Lukas’ group [87] revealed that a plasmonic structure that interacts with a fluorophore can lead to fluorescence enhancement depend on two factors. The first is the enhancement of the absorption rate, which is proportional to the intensity of the electric field component parallel to the absorption transition dipole at the fluorophore’s position. The second is the modification of the fluorophore’s quantum yield. Highly-ordered antenna dimers based on colloidal NPs can be aligned to the incident polarization at the dimer orientation, which will exhibit a significant fluorescence enhancement. This review mainly concentrates on the SEF-related latest development and practical applications.

Though the outperforming lithographic optical antennas have been used for the SEF phenomenon, their more widespread applications are limited because of their costly equipment, their time-consuming nature, and the laborious operations. Fluorescent-activating molecules can be tethered to the DNA sequence and the aforementioned DNA nanotechnology strategies can precisely place the fluorescent species next to one NP or in the gap between two NPs, which can largely amplify the fluorescence signal to the level of single-molecule emission by creating plasmonic nanoantennas that highly enhance the local fields [96,97]. In 2012, Acuna and co-works [98] initiated a typical study by attaching one or two AuNPs to DNA origami pillars and docking a single fluorescent dye in the vicinity of AuNP monomers and dimers of varying sizes. The authors have studied the dependence of the fluorescence intensity and lifetime of single dyes based on the nanoassemblies and achieved a fluorescence enhancement of up to 117-fold for the 100-nm dimers (Figure 7A). The mechanism of the electric field intensity for a monomer and dimer of 80-nm-diameter gold NPs and an interparticle spacing of 23 nm (dimer) was also discussed. They found that the fluorescence brightness enhancement originates from the highly enhanced local fields that are created by the plasmonic nanostructures and that it depends on how the excitation, as well as the radiative and nonradiative rates of the fluorescent dye, is influenced. This technique enables higher count rates in single-molecule applications and relaxes the requirements for single-molecule-compatible fluorescent dyes. Then, in 2015, researchers [99] continuously exploited a new generation of DNA origami nanoantennas by improving robustness, reducing the interparticle distance, and optimizing the quantum-yield improvement to achieve a maximal fluorescence enhancement of 5468 and a single-molecule detection at a 25-μM background fluorophore concentration (Figure 7B). This remarkable design and excellent optical property indicate the potential of the self-assembled nanoantennas for biosensing and emerging nano-biotechnological applications.

## 7. The Surface-Enhanced Raman Scattering of DNA-Directed AuNP Nanoarchitectures

In contrast to fluorescence, the Raman spectra that arise from the discrete vibrational energy levels of the molecules contain a chemical fingerprint of the measured molecule that is beneficial for detecting and identifying chemical compounds. However, the Raman scattering signals are very weak because most Raman-active molecules have a very small scattering cross-section. Therefore, it is more important and essential to enhance the Raman scattering signal. Fortunately, the Raman signal from molecules that are adsorbed into regions of high electromagnetic (EM) enhancement in the vicinity of the hotspot can be considerably enhanced when compared to the signal from free molecules, known as surface-enhanced Raman scattering (SERS) [100,101,102]. A common viewpoint is that the overall enhancement factor involves an electromagnetic enhancement factor E_EM_ associated with the plasmon-enhanced fields near the surface of a metal and a chemical enhancement factor E_CHEM_ that arises from a charge transfer between the surface and the molecule. Because of the weak chemical enhancement factor, we will only consider the electromagnetic contribution. Van Duyne’s group [103] demonstrated that the SERS electromagnetic enhancement factor is either the gap between the particles or, if the particles are fused, it is the radius of the curvature at the point of intersection. George C. Schatz’s group [104] discussed and focused on the modeling enhancement factor and asserted that the Raman signal is proportional to the product of the local electric field intensities at the incident frequency |E(ω)|^2^ and at the Stokes shifted frequency |E(ω′)|^2^. Since the Stokes shift is usually small in comparison to the LSPR width, it is usually taken as a reasonable approximation that the Raman enhancement is proportional to the fourth power of the local field at the incident frequency |E(ω)|^4^. They also used discrete dipole approximations (DDAs), the finite-element method (FEM), and the finite-difference-time-domain (FDTD) method to examine a variety of isolated and dimer structures, as well as the gap dependence of the dimer results (Figure 8). Therefore, plasmonic nanoantennas that emerged from metallic nanoparticles are capable of concentrating light into nanoscale volumes and creating a locally amplified EM field by coupling the localized surface plasmons of the nanoparticles [55,105,106]. The well-defined fabrication of metallic nanostructures with a high uniformity and engineering of the surface plasmon coupling is critical, able to amplify the Raman scatting signal of the analyte by several orders of magnitude and has great potential for a wide variety of biosensing and single-molecule applications.

Structural DNA nanotechnology enables the precise immobilization of single Raman probes in the hot-spot location which enhances the EM field, which has already been explored to fabricate those plasmonically coupled AuNP SERS nanoantennas at the nanoscale. Lan et al. [107] succeeded in investigating the particle size- and interparticle distance-dependance of the ensemble SERS properties through the DNA-directed self-assembly of AuNP dimers (Figure 9A). The Raman-active mercaptobenzoic acid molecule was anchored on the surface of the gold nanodimers. By tuning the constituent particle size and molecule length of the DNA bridge, they concluded that the size of the AuNP increases from 13 nm to 20 nm and 40 nm will result in the gradual increase of the SERS intensity. Furthermore, increasing the interparticle distance from 5 to 10 and 15 nm will dramatically decrease the SERS intensities. Apart from the DNA linker, the DNA origami-assembled AuNP dimers are more appropriate for the SERS study [108,109]. For example, Thacker et al. [109] employed an innovative origami design that allows for the accurate positioning of a 40-nm AuNP with strong plasmonic coupling from reliable 3.3 ± 1.0 nm gaps, which is one of the shortest controllable gaps that has yet been achieved with DNA origami assembly (Figure 9B). Their obtained SERS signal demonstrated the effectiveness of the AuNP dimer structures with local field enhancement factors that were up to seven orders of magnitude through the detection of a small number of dye molecules as well as short single-stranded DNA oligonucleotides. With this technique, other more complex AuNPs plasmonic nanostructures were also used to study the SERSs optical behavior. Zhao et al. [110] used the additional DNA linkers to organize the DNA origami-templated AuNP dimers into nanoribbons and revealed the significantly enhanced signals of the SERS of the AuNP dimers and AuNP-chains. Pilo-Pais et al. [111] selectively placed the four AuNPs on the corners of rectangular origami and subsequently enlarged them via solution-based metal deposition (Figure 9C). The resulting Raman-activate molecules deposited on the hotspot site of the tetramer AuNP assemblies was enhanced by at least 100 times compared to the individual nanoparticles. Although the gold nanospheres were readily arranged into a myriad of nano-patterns with the aid of a DNA template, it still remains a challenge to assemble other morphologically anisotropic NPs with real “sharp tips”, which could be done as plasmonic antennas amplify the Raman signal to a greater extent. Typical Au nanostars [112,113,114] and Au triangles [115] have sharp tips and even individually, such NPs can lead to an observable Raman scattering enhancement from the molecules at the tips where the field enhancement is the highest. If those sharp-tips-containing NPs could be exactly arranged into prescribed nanostructures, the strongest enhancement of the Raman signal could be recorded from the measurements on those hot spots. DNA origami is a reliable strategy to successfully anchor a single Raman probe between the gaps of the sharp-tips-containing nanoassemblies. In 2017, Tanwar et al. [56] created an assembly of Au nanostar dimers with a tunable interparticle gap and controlled stoichiometry assembled on dimerized rectangular DNA origami tied by branching staples (Figure 9D). The Raman reporter molecules were attached in the junction of the sharp tips of Au nanostar dimers. The obtained SERS enhancement factors reached values as high as 2 × 10^10^ and 8 × 10^9^ when their interparticle gaps were 7 and 13 nm, respectively. Recently, Ding’s group [57] successfully used a dimeric rectangle template strategy to assemble Au nanoprisms into a plasmonic bowtie configuration with an approximate 5-nm gap where the Raman probe was accurately positioned (Figure 9E). The apex-to-apex field coupling was able to enhance the collective local plasmonic behavior, which contributes to a mean enhancement factor of about 2.6 × 10^9^ and an electromagnetic field enhancement of about 2.3 × 10^2^.

## 8. Challenges and Perspectives

The tremendous progress made in specific plasmonic optical properties using DNA-guided nanostructures leads to broad applications such as sensing, energy transfer, and photonics, particularly at the single molecule detection level. For example, gold and silver nanoparticle dimers have already been employed as robust molecular rulers for the extended real-time monitoring of single-DNA hybridization events [116]. In addition, the interparticle junctions between DNA-based plasmonic nanostructures can be easily controlled when they are shorter than 10 nm, which is particularly advantageous over traditional fluorescent resonance energy transfer-based rulers and it is suitable for strong surface-enhanced Raman-scattering effects.

However, this technology still needs further improvements before it can live up to its extensive potential. For instance, DNA origami technique needs 7 kb of genomes of M13 as the primary source of the scaffold and plenty of DNA stable strands purchased from the DNA Company or solid phase synthesis with specialized equipment, which is costly or time-consuming. Moreover, larger and more complex superstructures can be potentially used for study of other unclear and special plasmonic optical properties, but the fabrication of plasmonic superstructures still remains a challenge because it is not easy to achieve higher-order DNA superstructures. To expand their size and complexity as DNA superstructures, there are also potential approaches that tried to tackle the problem via folding either a longer scaffold molecule or regulating the current origami as ‘super-tiles’ that can be linked together hierarchically to form larger superstructures. Even though larger origami nanoassemblies require the DNA linker staple to combine multiple origami, if those crucial staple sequences are degraded with DNase, it will largely reduce the efficiency of forming predesigned nanostructures. Maybe critical staples can be modified by introduction phosphoramidites to enhance their stability. In addition, high concentrations of magnesium ions are essential for maintaining the fold of the DNA origami structures, which greatly limit their application to magnesium-activated materials. How to keep the stability of DNA origami in low concentrations of magnesium ions still remains an elusive question.

Nevertheless, we believe that these troubles and problems can be solved with the further developments in nanosurface chemistry and DNA nanotechnology. Additionally, the DNA-based strategy for the assembly of plasmonic AuNPs can also be extended to a variety of materials involving quantum dots [117], carbon nanotubes [118,119], graphene [120], and other nanomaterials. By precisely controlling the interparticle distance, these heterogeneous material systems will open a new and bright chapter in nanoelectronics, nanophotonics, and nanosensors.

## 9. Conclusions

In this review, we provide a relatively comprehensive overview of the DNA-based assembly of gold nanostructures and their resultant unique optical properties including plasmonic extinction, plasmonic chirality, surface enhanced fluorescence, and surface-enhanced Raman scattering. Despite the great challenges because of the costing and stability, DNA linker and DNA origami structures still have a great potential for assembling a variety of nanoparticles independent of the geometry and elements. These resulting DNA composites enabled the efficient transport of signal reporters for photonics [121], optoelectronics [122], and nanomedicine [123].

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
