# Peer review of "DNA-Assisted Assembly of Gold Nanostructures and Their Induced Optical Properties"

_nanomaterials, 2018, doi:10.3390/nano8120994_

Round 1
Reviewer 1 Report
The review written by Ou et al. about DNA-assisted assembly of gold nanostructures summarizes well the state of art of this field. However, this work could be improved to better reach a wide auditorium:
-Authors should note the novelty of this review in comparison with other reviews related with self-assembled nanostructures based on DNA.
-Figure 1 should be improved or deleted because it does not complement the text.
- The review is considering optical properties of Au based nanostructures but general principles of optical phenomena involved should be also considered.
- The writing in the manuscript should be improved, there are several misspellings
-The challenge and perspective section is too short, authors should enlarge this section because is one the most relevant sections of a review
- The reviewer advices to include a new section about the chemistry of DNA assemblies
Author Response
1. Authors should note the novelty of this review in comparison with other reviews related with self-assembled nanostructures based on DNA.
Our response: Thanks for the reviewer’s suggestion. In the updated manuscript, we have highlighted our novelty of this review in comparison with other reviews related with self-assembled nanostructures based on DNA.
2. Figure 1 should be improved or deleted because it does not complement the text.
Our response: Thanks for the reviewer’s suggestion. Figure 1 showed DNA-linker-guided AuNP nanoarchitectures, which is the earliest method for assembling DNA-AuNPs composites. It has a great contribution to the following DNA nanotechnology. In the updated manuscript, we added many text to better illustrate the content of Figure 1, which makes it more reasonable and understandable.
3. The review is considering optical properties of Au based nanostructures but general principles of optical phenomena involved should be also considered.
Our response: Thanks for the reviewer’s suggestion. All general principles of optical phenomena were involved in separate section of our updated manuscript.
4. The writing in the manuscript should be improved, there are several misspellings
Our response: Thanks for the reviewer’s suggestion. We have carefully check our English writing and revised all misspellings.
5. The challenge and perspective section is too short, authors should enlarge this section because is one the most relevant sections of a review
Our response: Thanks for the reviewer’s suggestion. The challenge and perspective is so important for a review, we have enlarged this section in the updated manuscript.
6. The reviewer advices to include a new section about the chemistry of DNA assemblies
Our response: Thanks for the reviewer’s suggestion. We have added a new section “Structural DNA self-assembly nanotechnology” about the chemistry of DNA assemblies in our updated manuscript.
Reviewer 2 Report
Manuscript ID: nanomaterials-391777
Title: DNA-assisted assembly of gold nanostructures and their induced optical properties
Review report:
The review covers quite interesting area – DNA-based assembly of gold nanostructures and focuses namely to their optical properties, however there are some weaknesses which must be addressed before the article is acceptable for publication in Nanomaterials.
General comments to the authors
Authors are advised to revise the abstract. The sentence “Particularly, over the past…” the conclusion of the sentence is not linked to the preceding text, and the sentences act incompletely in its current form.
General problem of the manuscript is the usage of abbreviations. Any abbreviations used first time need to be explained properly, e.g (AuNPs, GNPs, CD).
Check the punctuation at line 31, 130.
The most serious deficiency is noticeable in absolute omitting the general principles of optical phenomena broadly discussed in the review article. I advise to the authors to incorporate short paragraph addressing this issue. Especially general principles of SERS, LSPR and others discussed mechanisms should be summarized in separate paragraph with emphasis to other particular phenomena such as red shift, blue shift, and orientation dependent response, regarding to mutual orientation of incident light (polarization) and nanostructures themselves. This will undoubtedly contribute to more comprehensible understanding of presented data by wide auditorium. Some useful information can be found in:
Krajcar, R.; Siegel, J.; Lyutakov, O.; Slepicka, P.; Svorcik, V. Optical response of anisotropic silver nanostructures on polarized light. Mater. Lett. 2014, 137, 72-74, 10.1016/j.matlet.2014.08.113.
Krajcar, R.; Siegel, J.; Slepicka, P.; Fitl, P.; Svorcik, V. Silver nanowires prepared on PET structured by laser irradiation. Mater. Lett. 2014, 117, 184-187, 10.1016/j.matlet.2013.11.112.
Barb, R.A.; Hrelescu, C.; Dong, L.; Heitz, J.; Siegel, J.; Slepicka, P.; Vosmanska, V.; Svorcik, V.; Magnus, B.; Marksteiner, R.; Schernthaner, M.; Groschner, K. Laser-induced periodic surface structures on polymers for formation of gold nanowires and activation of human cells. Appl. Phys. A 2014, 117, 295-300, 10.1007/s00339-013-8219-9.
Comments to the Figures
Figure 1. Description of the images is insufficient. Readers must easily understand individual processing steps in presented reaction sequences. Authors are advised to either add text in appropriate manner or split the figure on into more separate figures to enable more detail description.
Author Response
1. Authors are advised to revise the abstract. The sentence “Particularly, over the past…” the conclusion of the sentence is not linked to the preceding text, and the sentences act incompletely in its current form.
Our response: Thanks for the reviewer’s suggestion. We have revised this sentence in the abstract, which can logically and completely express and linked well to the preceding text.
2. General problem of the manuscript is the usage of abbreviations. Any abbreviations used first time need to be explained properly, e.g (AuNPs, GNPs, CD).
Our response: Thanks for the reviewer’s suggestion. We have explained properly all abbreviations when they are first used.
3. Check the punctuation at line 31, 130.
Our response: Thanks for the reviewer’s suggestion. We have revised the properly punctuation at line 31, 130.
4. The most serious deficiency is noticeable in absolute omitting the general principles of optical phenomena broadly discussed in the review article. I advise to the authors to incorporate short paragraph addressing this issue. Especially general principles of SERS, LSPR and others discussed mechanisms should be summarized in separate paragraph with emphasis to other particular phenomena such as red shift, blue shift, and orientation dependent response, regarding to mutual orientation of incident light (polarization) and nanostructures themselves. This will undoubtedly contribute to more comprehensible understanding of presented data by wide auditorium. Some useful information can be found in:
Krajcar, R.; Siegel, J.; Lyutakov, O.; Slepicka, P.; Svorcik, V. Optical response of anisotropic silver nanostructures on polarized light. Mater. Lett. 2014, 137, 72-74, 10.1016/j.matlet.2014.08.113.
Krajcar, R.; Siegel, J.; Slepicka, P.; Fitl, P.; Svorcik, V. Silver nanowires prepared on PET structured by laser irradiation. Mater. Lett. 2014, 117, 184-187, 10.1016/j.matlet.2013.11.112.
Barb, R.A.; Hrelescu, C.; Dong, L.; Heitz, J.; Siegel, J.; Slepicka, P.; Vosmanska, V.; Svorcik, V.; Magnus, B.; Marksteiner, R.; Schernthaner, M.; Groschner, K. Laser-induced periodic surface structures on polymers for formation of gold nanowires and activation of human cells. Appl. Phys. A 2014, 117, 295-300, 10.1007/s00339-013-8219-9.
Our response: Thanks for the reviewer’s suggestion. With the aid of the recommended three papers and other related works, we have added many text and figures to explain general principles of LSPR, chirality, SEF, SERS in our updated manuscript. We also cited the three useful papers in our updated manuscript.
Comments to the Figures
5. Figure 1. Description of the images is insufficient. Readers must easily understand individual processing steps in presented reaction sequences. Authors are advised to either add text in appropriate manner or split the figure on into more separate figures to enable more detail description.
Our response: Thanks for the reviewer’s suggestion. We have added separate text to descript figure 1 more detail in each work.
Reviewer 3 Report
Excellent and extensive review work. Up to date and comprehensive. Not much scientifical content to discuss from my part.
Author Response
Thank you very much for your approval.
Round 2
Reviewer 2 Report
Revission is weel done. Accept in current revised form.